# Role of Necroptosis in Intervertebral Disc Degeneration

**DOI:** 10.3390/ijms242015292

**Published:** 2023-10-18

**Authors:** Md Abdul Khaleque, Jae-Hoon Kim, Byung-Jun Hwang, Jin-Kyu Kang, Meiling Quan, Young-Yul Kim

**Affiliations:** Department of Orthopedic Surgery, Daejeon St. Mary’s Hospital, College of Medicine, The Catholic University of Korea, Daejeon 34943, Republic of Korea; abdulkhaleque.dream@gmail.com (M.A.K.); superbdoc@daum.net (J.-H.K.); soybeankk@naver.com (B.-J.H.); gamaer21@naver.com (J.-K.K.); mlquan225@beihua.edu.cn (M.Q.)

**Keywords:** necroptosis, RIPK-1, RIPK-3, MLKL, IVDD, PCD

## Abstract

Apoptosis has historically been considered the primary form of programmed cell death (PCD) and is responsible for regulating cellular processes during development, homeostasis, and disease. Conversely, necrosis was considered uncontrolled and unregulated. However, recent evidence has unveiled the significance of necroptosis, a regulated form of necrosis, as an important mechanism of PCD alongside apoptosis. The activation of necroptosis leads to cellular membrane disruption, inflammation, and vascularization. This process is crucial in various pathological conditions, including intervertebral disc degeneration (IVDD), neurodegeneration, inflammatory diseases, multiple cancers, and kidney injury. In recent years, extensive research efforts have shed light on the molecular regulation of the necroptotic pathway. Various stimuli trigger necroptosis, and its regulation involves the activation of specific proteins such as receptor-interacting protein kinase 1 (RIPK1), RIPK3, and the mixed lineage kinase domain-like (MLKL) pseudokinase. Understanding the intricate mechanisms governing necroptosis holds great promise for developing novel therapeutic interventions targeting necroptosis-associated IVDD. The objective of this review is to contribute to the growing body of scientific knowledge in this area by providing a comprehensive overview of necroptosis and its association with IVDD. Ultimately, these understandings will allow the development of innovative drugs that can modulate the necroptotic pathway, offering new therapeutic avenues for individuals suffering from necroptosis.

## 1. Introduction

IVDD is a common cause of lower back pain and disability worldwide [1,2,3,4,5]. The intervertebral disc (IVD) consists of a central nucleus pulposus (NP) surrounded by the annulus fibrosus (AF) and cartilaginous endplates [5,6,7,8,9]. The disc’s cell grows and repairs slowly due to a lack of blood vessels [10,11]. Factors such as mechanical stress, injury, smoking, and aging reduce blood supply, which can further impact nutrient delivery, potentially triggering IVDD [7,10]. From the early to the mid-1990s, it was thought that there were two types of cell death: one was PCD, which included apoptosis and autophagy, and the other was an unprogrammed, passive form of cell death called necrosis [12,13]. Specific morphological and biochemical properties characterize each of these types. Apoptosis is characterized by small cell size, cell membrane blebbing, pseudopod retraction, and nuclear fragmentation accompanied by chromatin condensation [14,15,16]. Autophagy is characterized by the accumulation of double membrane-covered vacuoles in the cytosol or organelles, as well as the redistribution of the light chain 3 (LC3) proteins to the autophagosome membrane [11,16,17]. Necrosis is a form of passive, non-physiological, and non-adjustable cell death resulting from accidental and acute damage to cells [12,18,19]. The biochemical components of these cell death pathways differ significantly. Apoptosis is regulated by the B cell lymphoma-2 (BCL-2) family of anti-and pro-apoptotic proteins, with hundreds of caspases involved in the process [20,21]. Autophagy is regulated by the autophagy-related (ATG) protein family, particularly ATG5 and ATG7, with LC3-I, LC3-II, and p62 [7,22,23]. For decades, apoptosis and autophagic cell death have been viewed as the only precisely regulated, direct forms of cell death. In contrast, necrosis involves the release of cellular contents, the swelling of organelles, and the rupturing of the plasma membrane, constituting an unregulated, indirect form of cell death. However, recent research has shed light on cell death processes, revealing that necrosis can also occur in a highly regulated manner and operate under genetic controls, a phenomenon named necroptosis. The formation of necroptosis largely depends on the RIPK1, RIPK3, and MLKL signaling pathways, which have recently received significant attention in IVDD research [13,24].

Some researchers use the terms “necroptosis” and “necrosis” interchangeably. However, it is crucial to note that a significant distinction exists between these processes, despite their shared morphological features. Both involve cellular swelling, the release of intracellular contents, and the inducing of a pro-inflammatory response [25,26]. However, necrosis is unregulated and is linked to conditions like IR injury, neurodegeneration, and infarction. Cellular breakdown leads to inflammation due to disruptions caused by calcium and ROS [27,28]. This gives rise to mitochondrial dysfunction, the disruption of the ion balance, and the loss of membrane integrity [25]. On the other hand, decades ago, necroptosis emerged as a regulated cell death mode that resembles necrosis but is functionally similar to apoptosis. Necroptosis requires a necrosome with RIPK1, RIPK3, and MLKL pseudokinases, triggered by factors like the tumor necrosis factor (TNF), Fas, TNF-related apoptosis-inducing ligand receptor (TRAIL-R), Toll-like receptors (TLRs), interferon gamma (IFN-γ), infections, and drugs(see Figure 1) [25,27,29,30].

Thus, the necroptosis mechanisms are briefly summarized. The accumulated evidence favoring the role of necroptosis in IVDD is given here.

## 2. History of Necroptosis

Rudolf Virchow, a renowned pathologist of the mid-19th century, made a significant observation by recognizing necrosis as a type of cell death associated with the disease. During the last century, apoptosis emerged as a distinct process characterized by molecular mechanisms orchestrating PCD. The prevailing belief was that necrosis and apoptosis are fundamentally different. This was primarily due to the notion that necrosis lacks underlying cellular events. However, recent investigations have demonstrated the existence of specific molecular pathways that induce necrotic cell death (NCD), now referred to as “necroptosis” [27]. In 2005, Degterev et al. introduced the term “necroptosis”, denoting a form of necrosis regulated by RIPK1 [31]. This nomenclature aligns with recent studies, emphasizing necroptosis dependence on RIPK3 [32]. Although the stimulation of the Fas/TNFR receptor family traditionally triggers a canonical extrinsic apoptosis pathway without intracellular apoptotic signaling, it has been discovered that it can also activate a shared non-apoptotic pathway, now known as necroptosis [33].

In this article, we will discuss the role of necroptosis in IVDD. It is worth noting that while some researchers use the term “necroptosis” to describe any active necroptosis, we use it based on recent recommendations.

## 3. Molecular Mechanisms of the Necroptosis Pathway

Necroptosis is a form of PCD triggered by death receptor necrosis factors such as tumor necrosis factor receptor 1 (TNFR1), tumor necrosis factor receptor 2 (TNFR2), and ligands like TNF-α and Fas-binding ligands (FasL, inhibiting the apoptotic pathway [27,34]. The TNF-α binding to TNFR1 on the cell membrane triggers a conformational change in TNFR1. This change leads to the recruitment of several proteins, including the TNFR1-associated death domain (TRADD), RIPK1, the TNFR-associated factor 2/5 (TRAF2/5), a cellular inhibitor of apoptosis protein (cIAP1and cIAP2), and the ubiquitination complex to the TNFR complex [28,34]. After the formation of TNFR complex I, an intricate interplay of signaling pathways is orchestrated, involving ubiquitination and phosphorylation processes that ultimately determine the cell’s fate: either survival or death [35,36,37]. The TNFR complex I emerges as a critical factor for influencing a multitude of downstream pathways [6]. Notably, when RIPK1 is polyubiquitinated within TNFR complex I, it serves as a scaffold for the recruitment of transforming growth factor-β activated kinase 1 (TAK1), TAK1-binding protein 2 (TAB2), and TAB3. Together, this complex (TAK1-TAB2-TAB3) triggers the activation of the nuclear factor kappa B (NF-κB) signaling pathway. The activation of NF-κB promotes cell survival and facilitates the expression of pro-survival genes. Conversely, the deubiquitination and subsequent release of RIPK1 from TNFR complex I stimulates the formation of the TNFR complex II [30,38]. The TNFR complex II serves as a platform for initiating the PCD processes. Upon thedissociation of TRADD and the Fas-associated death domain (FADD) from complex I, complex IIa comprises pro-caspase-8, RIPK1, and FADD. Complex IIa plays a pivotal role in promoting the activation of caspase-8. Activated caspase-8, in turn, triggers apoptosis by activating caspase-3, a key effector of apoptotic cell death [36,37,39]. The assembly of complex IIb, which includes FADD, RIPK1, and pro-caspase-8, occurs independently of TRADD and can induce caspase-8-dependent apoptosis [40,41,42]. In instances where caspase-8 is inhibited, RIPK1 interacts with RIPK3 through a specific region called the RIPK homotypic interaction motif (RHIM). This interaction leads to the formation of complex IIc/necrosome. Complex IIc/necrosome serves as a functional amyloid protein complex essential for signal transduction and the activation of necrosis, a form of programmed necrotic cell death [40,43]. Moreover, TNFR complex IIc/necrosome is crucial in initiating necroptosis, another form of PCD. Once necroptosis is initiated, the necrosome facilitates the formation of a disulfide bond-dependent MLKL polymer. This process involves the activation of MLKL, a pseudokinase, through phosphorylation at specific sites such as threonine 357/serine 358 in humans or serine 345 (in mice) [33,44]. Consequently, the RIPK1-RIPK3-MLKL complex assembles. Subsequently, MLKL undergoes structural changes and translocates to the plasma membrane, where it induces cell rupture and the execution of necroptosis [40,45]. Upon binding to its ligand, other death receptors like Fas recruit a membrane-associated death complex consisting of FADD and caspase-8 [46]. This complex formation occurs in certain conditions where caspase-8 is absent but cellular inhibitors of apoptosis proteins (cIAP) are present, and Fas can also facilitate the assembly of the necrosome through a RIPK3-dependent mechanism [47]. The regulation of the necroptosis signaling pathway mainly revolves around three key molecules: RIPK1, RIPK3, and MLKL. The C-terminus of RIPK1 and RIPK3 possesses a distinctive protein–protein interaction domain called the RHIM domain, which facilitates the interaction between RIPK1 and RIPK3. This interaction is essential in the TNF-induced necroptosis pathway, where RIPK1 acts as a signaling molecule required for activatingRIPK3. However, RIPK1 is not a prerequisite for RIPK3 activation in other necroptosis pathways [43,48]. In addition to its role in the TNF-induced necroptosis pathway, RIPK3 can activate Ca^2+^/calmodulin-dependent protein kinase II (CaMK II), which, in turn, triggers the opening of the mitochondrial permeability transition pore (mPTP). This activation ultimately leads to necroptosis in cardiomyocytes [49]. Apart from being activated by RIPK1, RIPK3 can also be activated by other proteins that possess the RHIM domain, such as the TIR domain-containing adaptor protein (TRIF) and the DNA-dependent activator of interferon regulatory factors (DAI) [50]. This expands the range of activation mechanisms for RIPK3 in necroptosis. For example, Toll-like receptor 3 (TLR3) or TLR4 can directly activate the TRIF–RIPK3–MLKL pathway of necroptosis. This activation occurs through the synergistic interaction between the RHIM domain in TRIF and RIPK3. Thus, the TRIF–RIPK3–MLKL pathway is initiated, leading to necroptosis [51,52]. RIPK3 acts as a signal integration molecule that can be employed based on the demand for necroptosis. By interacting with other signal molecules containing the RHIM domain, RIPK3 can initiate diverse necroptosis pathways, resulting in distinct cellular outcomes. 

MLKL, considered a pseudokinase, undergoes activation through a sequence of events: kinase-like domain dimerization, self-assembly in the coiled-coil region, and MLKL oligomer formation. The phosphorylation of MLKL by RIPK3 drives kinase-like domain dimerization, while essential functional integrity relies on coiled-coil self-assembly [53,54]. This process is conserved among mammals, indicating a shared activation mechanism. Polymerized MLKL binds to lipids and disrupts the cell membrane causing necroptosis (See Figure 2) [27,34,40,55,56,57,58].

**Table 1 ijms-24-15292-t001:** Contributing factors and pathways involved in IVDD.

Serial No.	Contributing Factor	Pathway Involved	Outcomes	Target	Reference
1	Compression	Mitochondrial dysfunction and (ROS) reactive oxygen species	Compression led to a time-dependent decrease in (ATP) Adenosine triphosphate production and increased oxidative stress, resulting in mitochondrial membrane potential (MMP) loss and the promotion of mitochondrial integrity. Finally, mitochondrial dysfunction occurs.	NP cells	[11]
2	Compression	(HSP 90) heat shock protein	Compression induces HSP90, which, in turn, triggers necroptosis through the JNK pathway when caspase is absent or inhibited.	NPSC	[17]
3	Compression	RIPK1/RIPK3/MLKL	The RIPK1–RIPK3–MLKL complex assembles. Consequently, due to complex formation, MLKL undergoes structural changes and translocates to the plasma membrane, where it induces cell rupture and executes necroptosis.	NP cells	[22]
4	MyD88	Mitochondrial dysfunction and ROS	TLRs activate the MyD88 complex, which induces RIPK1, recruiting RIPK3 and MLKL to form a RIPK1–RIPK3–MLKL complex. This complex binds to the PGAM5 protein of the mitochondria, eventually generating ROS and opening the potential transition pore(PTP), increasing mitochondrial dysfunction, and finally, necroptosis is initiated.	NP cells	[58]
5	TNF-α or IL-1β	RIPK1/RIPK3/MLKL, mitochondrial dysfunction, and ROS	Inflammatory cytokines increase mitochondrial dysfunction through ROS generation and FADD-mediated RIPK1/RIPK3/MLKL, producing necrosomes via MLKL octamers, which cause membrane rupture and execute necroptosis.	NP cells	[59]
6	Compression	Drp1	Compression induces Drp-1, which translocates to the mitochondria and increases the P53 protein. These factors form a complex that produces ROS and opens the PTP, ultimately leading to the initiation of necroptosis.	NP cells	[60]
7	Compression	Endoplasmic reticulum stress and ER-mitochondrial Ca^2+^	Compression induces ER swelling, which increases Ca^2+^ from ER-to-mitochondria transfer via specific proteins, and the subsequent activation of the PARP–AIF pathway, with ROS accumulation as a trigger for ERS and Ca^2+^ signaling.	NP cells	[61]
8	Hydrogen peroxide	Mitochondrial dysfunction, and PARP–AIF pathway	Hydrogen peroxide induces ROS generation, leading to mitochondrial dysfunction by binding to mitochondrial proteins and activating the PARP–AIF pathway, ultimately forming necroptosis.	NP cells	[62]

## 4. Role of Necroptosis in IVDD

### 4.1. Compression-Induced RIPK1/RIPK3/MLKL-Mediated Necroptosis

Compression-induced IVDD is associated with necrosis in nucleus pulposus (NP) cells [5,63]. Prolonged compression leads to cell detachment and necrotic changes. Yurube, T., et al. verified this phenomenon using compounds such as Nec-1(Necrostatin-1), GSK’872, and NSA (Necrosulfonamide), which have shown promising results in reducing necrosis and preserving cell viability in compressed NP cells. Specifically, Nec-1 treatment decreased cell death and improved viability. Ultrastructural analysis further supported the role of Nec-1 and NSA in mitigating necrotic changes induced by compression. Furthermore, it has been reported that the key proteins associated with necroptosis, including RIPK1, RIPK3, and MLKL, showed increased expression with prolonged compression. Additionally, some researchers have suggested that the phosphorylated forms of RIPK1 and RIPK3 play a role in the process. Moreover, compression-induced damage to mitochondria was evident, as indicated by the loss of MMP in NP cells. Treatment with Nec-1 or CsA (cyclosporine A), an inhibitor of cyclophilin D, showed protective effects by preventing MMP loss and promoting mitochondrial integrity. Compression also led to a time-dependent decrease in ATP production and increased oxidative stress. Nec-1 and CsA treatments alleviated these effects, reducing ROS levels and maintaining antioxidant activity. Moreover, SiRIPK1 treatment, targeting RIPK1 sequences, worsened mitochondrial dysfunction and oxidative stress, highlighting the crucial role of RIPK1 in maintaining mitochondrial function and protecting against oxidative stress in compressed NP cells. This finding strongly supports the significant contribution of necroptosis to NP cell death induced by compression (See Table 1 and Figure 3) [13,24].

### 4.2. MyD88-Induced Necroptosis Mediated IVDD

Myeloid differentiation primary response 88(MyD88) is a protein that, in humans, is encoded by the MyD88 gene and is a signaling molecule in innate immunity [59]. MyD88 plays a significant role in the necroptosis of NP cells during IVDD. Fan, H., et al. demonstrated that certain levels of RIP3 and MLKL increased in NP and AF cells of degenerated discs compared to normal discs. Furthermore, a positive correlation was also identified between MyD88 gene expression and the severity of IVDD. The MyD88 inhibitor was used for in vitro experiments to explore the role of MyD88 in necroptosis. It rescued decreased cell viability, ATP levels, and increased ROS levels induced via necroptosis induction, while having no effect in the absence of necroptosis induction. Flow cytometry data verified that the MyD88 inhibitor decreased necrosis in NP cells. Transmission electron microscopy revealed that the MyD88 inhibitor partially protected mitochondrial ultrastructure from TLZ (talazoparib)-induced damage, indicating its involvement in mitigating necroptosis by preserving mitochondrial integrity. Overall, the findings suggest that MyD88 signaling plays a pivotal role in NP cell necroptosis during IVDD (See Table 1 and Figure 3) [64].

### 4.3. Inflammatory Stimulation Induces Necroptosis and Mediates NP Cell Death

Inflammation is a critical defense mechanism against infection or threats, but its dysregulation results in inflammatory diseases. Successful treatments have focused on pro-inflammatory cytokines, particularly TNF, which directly stimulate genes responsible for inflammatory responses. Additionally, TNF and similar signals can directly induce caspase-independent cell death; known as necroptosis, further promoting inflammation [57,65]. Cao, C., et al. demonstrated that treatment with pro-inflammatory factors TNF-α or IL-1β led to the increased expression of necroptosis-associated molecules, including RIPK1, phosphorylated RIPK1(p-RIPK1), RIPK3, phosphorylated RIPK3 (p-RIPK3), MLKL, and phosphorylated MLKL (p-MLKL), in NP cells. They verified the involvement of necroptosis in cell death using Nec-1, which effectively improved cell activity, inhibited LDH (Lactate dehydrogenase) release (a marker of cell death), and reduced PI positivity (indicating cell death). Further specific roles of RIPK1, RIPK3, and MLKL were investigated using small interfering RNA (SiRNA). Down-regulating RIPK1 activity with SiRIPK1-aggravated cell death, while down-regulating RIPK3 and MLKL with SiRIPK3 and SiMLKL significantly inhibited inflammation-induced NP cell death and increased cell activity. Furthermore, MMP loss was noted in NP cells treated with TNF-α or IL-1β, but treatment with Nec-1, GSK872, or NSA effectively prevented MMP loss, indicating their protective effects against necroptosis. Additionally, cells were exposed to TNF-α or IL-1β, increasing ROS levels. However, after using compounds such as Nec-1, GSK872, or NSA, ROS generation was significantly reduced. This proved that inflammation-induced NP cell death involves necroptosis mediated by RIPK1, RIPK3, and MLKL (See Table 1 and Figure 3) [60].

### 4.4. Compression-Induced HSP90Mediated the Necrotic Cell Death of NPSCs (Nucleus Pulposus-Derived Stem/Progenitor Cells)

HSP90, a molecular chaperone, is crucial in preserving cellular functionality, stability, and viability when exposed to transformative stimuli [66]. Hu, B., et al. mentioned that compression led to a time-dependent reduction in NPSCs viability and caused cellular damage and necrosis. The observational data demonstrated the heightened expression levels of RIPK1, p-RIPK1, RIPK3, p-RIPK3, MLKL, and p-MLKL. Following the administration of inhibitors targeting these proteins, a decrease in the proportion of necrotic cells was observed, and the preservation of NPSCs viability suggests a reduction in necrotic cell death. Furthermore, the elevated expression of HSP90 was found in degenerated disc tissues, and inhibiting HSP90 with BIIB021 attenuated the loss of NPSCs viability caused by compression. The inhibition of HSP90 increased the expression of HSP70, which played a protective role by inhibiting JNK activity. This evidence revealed an association between HSP90 and necroptosis, as inhibiting HSP90 reduced necrotic cell death and the expression of key necroptosis-related proteins. The inhibition of HSP90 effectively protected NPSCs from compression-induced death, suppressed necroptosis, and attenuated the depletion of NPSCs in the IVD tissue.This study demonstrates that inhibiting HSP90 plays a protective role in the compression-induced necroptosis of NPSCs (See Table 1 and Figure 3) [19].

### 4.5. Compression-Induced Drp1 (Dynamin-Related Protein) Mediated Programmed Necrosis in IVDs

Drp1 is essential for mitochondrial and peroxisomal fission, but it has both membrane constricting and severing abilities [61]. However, a recent study suggested that Drp1 has a role in the compression-induced programmed necrosis of NP cells in intervertebral discs. Drp1 expression increased in NP cells and was translocated to the mitochondria in response to compression. Furthermore, silencing Drp1 expression or inhibiting Drp1 with mitochondrial division inhibitor-1(mdivi-1) improved cell viability and reduced necrotic cell death in compressed NP cells. Transmission electron microscopy (TEM) analysis also revealed that compression induced necrotic morphological changes in NP cells, alleviated by mdivi-1 treatment. Additionally, the release of LDH and high mobility group box 1 (HMGB1), markers of necrotic cell death, increased in compressed NP cells but were inhibited by mdivi-1 treatment. Moreover, compression increased p53 expression in the mitochondria of NP cells, and treatment with mdivi-1 inhibited its mitochondrial translocation. The inhibition of p53 using a specific inhibitor, namely pifithrinm (PFTm), reduced necrotic cell death in compressed NP cells. Additionally, AIF was identified, which translocates from mitochondria to the nucleus during compression, and this translocation was reversed via mdivi-1 treatment. The silencing of AIF expression prevented necrotic cell death in compressed NP cells. This result demonstrates that Drp1 mediates compression-induced programmed necrosis in NP cells (See Table 1 and Figure 3) [62].

### 4.6. ROS Regulate (ERS) Endoplasmic Reticulum Stress and ER-Mitochondrial Ca^2+^Crosstalk to Promote the Programmed Necrosis of Rat NP Cells during Compression

The ER plays a critical role in cellular activities and survival. Disruptions in ER function cause the buildup of unfolded proteins, activating the unfolded protein response to restore normal ER function. If the adaptive response fails, it triggers apoptosis or PCD [37,67]. However recent studies have mentioned that ERS and calcium (Ca^2+^) signaling are involved in compression-induced programmed necrosis in NP cells mediated by IVDD. ERS-related proteins expression, such as C/EBP homologous protein [39], glucose-regulated protein 78 (GRP78), protein kinase R (PKR), and endoplasmic reticulum kinase (pPERK), increased in a time-dependent manner in degenerated discs, Furthermore, the mRNA levels of CHOP, GRP78, and PERK increased in compressed NP cells. Electron microscopy revealed ER swelling and ER-mitochondria interaction under compression. The inhibition of ERS using 4-phenylbutyric acid (4-PBA) attenuated compression-induced programmed necrosis, as evidenced by decreased cell death and the reduced release of necrotic markers (HMGB1 and LDH) and inhibited Ca^2+^ accumulation in mitochondria. Furthermore, this study demonstrated the involvement of IP3R-mediated ER-to-mitochondria Ca^2+^ transfer. Silencing the key proteins reduced Ca^2+^ levels and attenuated programmed necrosis. The inhibition of mitochondrial Ca^2+^ uptake preserved mitochondrial function and attenuated cell death. Compression also activated the PARP-AIF pathway, which was modulated by Ca^2+^ signaling. Scavenging ROS attenuated programmed necrosis and ERS activation. In summary, the study suggests that compression-induced programmed necrosis of NP cells involves ERS, ER-to-mitochondria Ca^2+^ transfer via specific proteins, and the subsequent activation of the PARP–AIF pathway. Additionally, it proposes that ROS accumulation acts as a trigger for ERS and Ca^2+^ signaling (See Table 1 and Figure 3) [68].

### 4.7. Hydrogen Peroxide Induces Programmed Necrosis in Rat NP Cells via the RIPK1/RIPK3–PARP–AIF Pathway

Zhao, L., et al. demonstrated that H_2_O_2_ induces programmed necrosis in rat NP cells through the RIPK1/RIPK3–PARP–AIF pathway. H_2_O_2_ exposure caused cell death, but pretreatment with the necroptosis inhibitor Nec-1 reduced cell death and improved cell viability. H_2_O_2_ increased RIPK1 and RIPK3 expression, while Nec-1 effectively reduced their levels. Ultrastructural changes associated with H_2_O_2_-induced necrosis were improved by Nec-1. Furthermore, RIPK3 knockdown reduces cell death, while RIPK1 knockdown exacerbates it, confirming their roles in cell death regulation. H_2_O_2_ also caused an increase in ROS, mitochondrial dysfunction, and PARP–AIF pathway activation, all of which were mitigated by Nec-1 or RIPK3 knockdown. The inhibition of the PARP–AIF pathway improved cell viability and reduced cell death. This evidence proved that H_2_O_2_ induces programmed necrosis in rat NP cells through the RIPK1/RIPK3–PARP–AIF pathway and suggested potential therapeutic targets for intervertebral disc degeneration (See Table 1 and Figure 3) [24,69].

### 4.8. Crosstalk with Other Regulated Cell Death RCD

The IVDD is widely recognized to be influenced by the death of NP cells, which can be exacerbated by compression [70,71]. Necroptosis, a newly identified form of PCD, has been found to play a significant role in the death of NP cells induced by compression [13,24]. Previous research has indicated that autophagy, a cellular degradation process, is downstream of necroptosis. However, there are three prevailing perspectives regarding the interaction between necroptosis and autophagy [72]. It has been suggested autophagy activation up-regulates necroptosis, reducing autophagy activation is a consequence of necroptosis [73]. Moreover, it has been reported that a reciprocal conversion between necroptosis and apoptosis may exist, and the simultaneous inhibition of necroptosis and apoptosis leads to the more efficient survival of NP cells compared to the inhibition of necroptosis alone. This phenomenon could be closely associated with the mitochondrial dysfunction–oxidative stress pathway [26,73]. Furthermore, the literature also suggests that blocking apoptosis can either enhance or decrease necroptosis, implying that inhibiting apoptosis could potentiate or attenuate the progression toward necroptosis [72]. Similarly, the inhibition of necroptosis has been reported to promote or reduce apoptosis [74]. Therefore, a comprehensive understanding of the “crosstalk effect” between necroptosis and apoptosis is necessary to identify effective intervention targets for preventing NP cell death [73].

## 5. Potential Treatment to Inhibit Necroptosis

Necroptosis has been identified as a potentially crucial contributing factor to the development of IVDD. Given the pivotal role that specific groups of protein molecules play in the necroptotic process, various anti-necroptosis strategies have been explored in animal models as potential treatments for IVDD. Notably, substances such as Nec-1, GSK’872, and NSA have exhibited significant inhibitory effects on necroptosis, thereby showing promise in preventing IVDD [75]. Our discussion delves into the mechanisms by which these inhibitors modulate necroptosis, aiming to identify novel agents with specific targeting capabilities for necroptotic pathways. In this context, the elucidation of necroptosis inhibitors holds the potential to pave the way for innovative and targeted treatment strategies aimed at mitigating IVDD.

### 5.1. Potential Inhibitors to the Target of RIPK1

In 2005, Degterev and collaborators confirmed the inhibitory potential of Nec-1, a specialized, low-molecular-weight necroptosis inhibitor, against TNF-induced necrosis in human monocytic U937 cells [31]. Subsequently, they validated its ability to suppress RIPK1 kinase activity [76]. Later studies led to the development of Nec-1s, an advanced necroptosis inhibitor with improved plasma stability and enhanced specificity for RIPK in various kinases. Importantly, it exhibited lower toxicity compared to Nec-1 [77]. Despite the emergence of multiple necroptosis inhibitors with varying structures, such as Nec-1, Nec-3, Nec-4, Nec-5, and Nec-7 [78], their specific roles in IVDD necessitate further exploration. Additionally, the potential applications of newer RIPK1 inhibitors like compound 56 (RIPA-56), VX-680, and MK-0457, which have demonstrated efficacy in various diseases, warrant deeper investigation within the context of IVDD [79]. In conclusion, targeting RIPK1 kinase shows promise as a novel approach for the development of therapeutic drugs for IVDD (See Table 2).

### 5.2. Potential Inhibitors to the Target of RIPK3

Among the three RIPK3 kinase inhibitors, namely GSK’840, GSK’843, and GSK’872, it is worth noting that higher concentrations of GSK’843 and GSK’872 have demonstrated the potential to promote RIP1-dependent apoptosis induced by TNF and the activation of caspase 8 [80]. However, GSK’840 exhibits a superior specificity profile. It is important to mention that GSK’840 lacks the ability to effectively inhibit murine RIPK3, making it less suitable for assessment in murine models. Dabrafenib, a B-Raf inhibitor, stands out as the only type I RIPK3 inhibitor that has received clinical approval for use [81]. It operates by attenuating the phosphorylation of MLKL viaRIPK3, thereby disrupting the RIPK3/MLKL interaction [82,83]. On a different note, HS-1371 has emerged as a novel type of II RIPK3 inhibitor, interacting with the ATP-binding pocket of RIPK3 [84]. However, it is important to acknowledge that recent clinical trial failures targeting RIPK3 in various diseases have raised questions about its suitability as an ideal target for blocking necroptosis (See Table 2).

### 5.3. Potential Inhibitors to the Target of MLKL

NSA is the first reported compound for inhibiting MLKL, serves as a specific downstream target for RIPK3 through identifying MLKL [33]. While NSA can prevent necroptosis in human cells, it is ineffective in murine cells due to its alkylation of Cys86 in human MLKL, which is absent in murine MLKL, making it unsuitable for murine preclinical models. IVDD is closely linked to inflammation, oxidative stress, and cell death. Zhang et al. demonstrated that NSA, as an MLKL inhibitor, prevented NP degradation [12]. They also observed significant reductions in inflammatory factors such as MMPs, IL-6, and TNF-α following NSA treatment. This suggests that inhibiting the protective effect of necroptosis is likely due to its ability to suppress inflammation and oxidative stress. Additionally, a new class of MLKL inhibitors that target the MLKL pseudokinase domain, including compound 1 (GW806742X or SYN-1215), has gained attention [85]. Thioredoxin-1 (Trx1), an oxidoreductase, can bind to MLKL and maintain it in a reduced state under normal conditions, thus preventing MLKL disulfide bond formation and polymerization and thereby blocking necroptosis [86]. However, drugs that up-regulate Trx1 are still under investigation. MLKL is a crucial target for inhibiting necroptosis (See Table 2).

### 5.4. Potential Inhibitors to the Target of ER Stress

Tauroursodeoxycholic acid (TUDCA) is a kind of hydrophilic bile acid, which could protect cells from death byinhibiting ER stress. TUDCA has been identified as an inhibitor of necroptosis and could be a potential therapeutic option for conditions involving necroptosis, such as IVDD [18] (See Table 2).

As the understanding of necroptosis progresses, it is becoming increasingly important to elucidate the regulatory mechanisms involved. While numerous posttranslational regulatory molecules have been identified, certain modifications, such as the ubiquitination of RIPK3 and MLKL within the necrosome, still require further investigation [87]. Recent studies have also unveiled intricate crosstalk between necroptosis signaling and other cellular pathways, suggesting complex interactions between necroptosis and diverse signaling networks [29].

In a recent study, combination treatment with necroptosis inhibitor Nec-1 and apoptosis inhibitor Z-VAD was shown to effectively block the opening of the mPTP and prevent the loss of mitochondrial membrane potential in NP cells. This protective effect of the combined treatment could be attributed to the restoration of mitochondrial function, further highlighting the significance of mitochondrial dynamics in the context of necroptosis [73,88].

## 6. Conclusions and Future Prospects

IVDD represents a global health concern due to its substantial disability impact. Growing empirical evidence underscores the pivotal role of necroptosis in the pathophysiological processes associated with IVDD, thus rendering it a promising therapeutic target for the condition. In our review, we provide a concise summary of how necroptosis occurs and discuss the mechanisms behind it. We also explore the connection between necroptosis and IVDD, as well as the potential for inhibiting necroptosis as a therapy for IVDD. However, it is important to note that research in this area is still in its early stages, and many questions remain unanswered.

Key markers involved in necroptosis-mediated NP cell death include RIPK1, RIPK3, and MLKL [24]. Additionally, mitochondrial dysfunction, MDy88 protein, inflammatory cytokines (e.g., TNF-αorIL-1β), HSP90, Drp1, ERS, calcium signaling, and the RIP1/RIP3-PARP-AIF pathway have been identified as significant contributors to IVDD [19,59,60,61,62,68,69]. Although there is substantial evidence indicating an increase in necroptosis-related markers in experimental conditions, like compression, and their inhibition by specific inhibitors, the exact pathway responsible for the up-regulation of these necroptotic markers remains elusive. Moreover, it has been suggested that hypoxia induces necroptosis under the conditions of nutrient deprivation through mitochondrial dysfunction as a potential mechanism that can be investigated in the context of IVDD [89]. There is contradictory evidence regarding the interaction between apoptosis and necroptosis, with some studies indicating that the inhibition of apoptosis enhances necroptosis while others showed a decrease [73,88]. Further extensive research is needed to elucidate the underlying mechanisms behind this phenomenon. Hu et al. highlighted that HSP90 triggers cell death through necroptosis via the JNK pathway, but the mechanisms driving necroptosis-mediated cell death remain unclear, creating a knowledge gap. Furthermore, it has been proposed that autophagy attenuates necroptosis, while others suggested autophagy as a downstream consequence of necroptosis [73]. The interplay between necroptosis and autophagy also can be investigated. The involvement of ERS and calcium signaling pathways in necroptosis-mediated IVDD has been identified [68], although limited research has been conducted in this area. More extensive studies are needed to better understand the mechanisms underlying the necroptosis-mediated IVD.

Therefore, it becomes imperative to delve deeper into the links between necroptosis and these various cell death mechanisms and their distinct roles within the context of IVDD. IVDD is a complex condition, and it is increasingly evident that it may not be solely driven by a single form of cell death. This insight suggests that therapies designed to target multiple pathways of cell death simultaneously might hold greater promise for effectively managing IVDD. In conclusion, gaining a more profound comprehension of the involvement of necroptosis in IVDD would significantly enhance the development of more potent and efficacious therapeutic approaches for this condition.

## Figures and Tables

**Figure 1 ijms-24-15292-f001:**
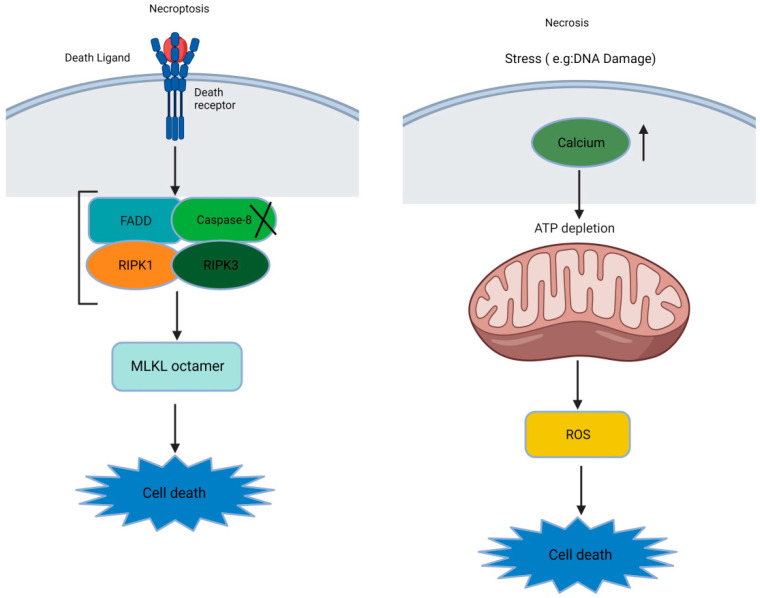
The basic difference between necroptosis and necrosis ([X: Inhibition or inactivation], [↑: increase], and [↓: downstream sequential events]).

**Figure 2 ijms-24-15292-f002:**
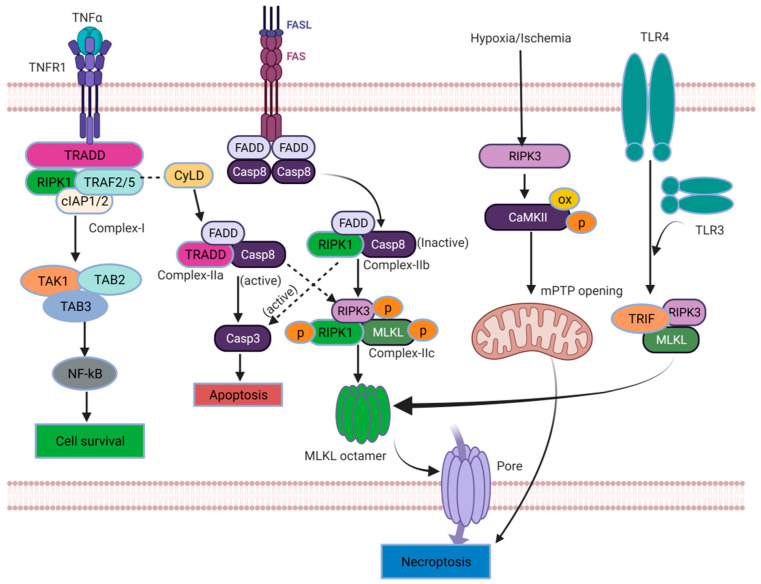
Molecular mechanisms of necroptosis ([OX: oxidative stres], [→: Primary events], [
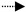
: Secondary events]).

**Figure 3 ijms-24-15292-f003:**
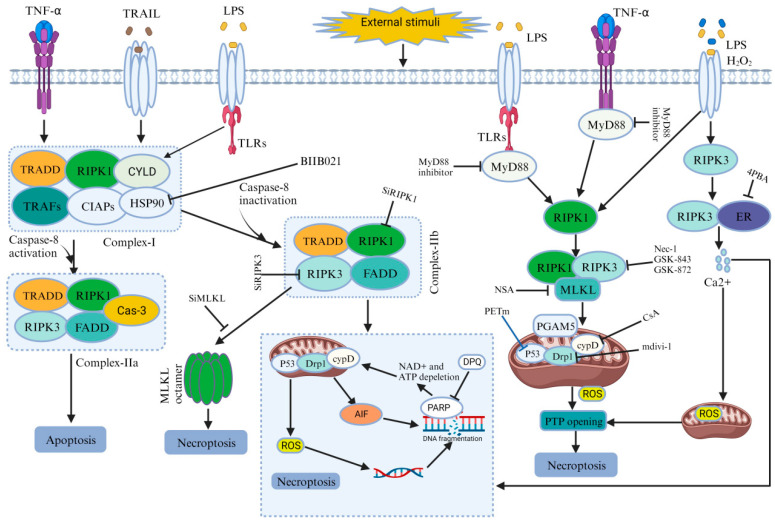
Role of necroptosis in IVDD.

**Table 2 ijms-24-15292-t002:** Potential targets associated with necroptosis and their specific inhibitors.

Name of Inhibitors	Target	Specific Functions
Nec-1, Nec-3, Nec-4, Nec-5, and Nec-7, furo[2,3-d]pyrimidines, and GSK’963	RIPK1	Suppresses RIPK1 kinase activity
RIPA-56, VX-680, and MK-0457	RIPK1	Requires further investigation
GSK’840, GSK’843, and GSK’872	RIPK3	Suppresses RIPK3 kinase
Dabrafenib	RIPK3	Suppresses RIPK3 kinase
HS-1371	ATP-binding pocket of RIPK3	Suppresses RIPK3 kinase
NSA	MLK	Inhibits MLKL
Thioredoxin-1 (Trx1)	MLKL	RMLKL disulfide bond formation and polymerization
Tauroursodeoxycholic acid (TUDCA)	ER	Inhibits ER stress ROS
Hydroxyanisole	Cytoplasm	Blocks ROS accumulation
Diphenyleneiodonium(DPI)	Mitochondria	NADPH oxidase inhibitor

## Data Availability

The authors confirm that the data supporting the findings of this study are available within the article.

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
