# Peer review of "Role of Necroptosis in Intervertebral Disc Degeneration"

_ijms, 2023, doi:10.3390/ijms242015292_

Round 1

Reviewer 1 Report

Comments and Suggestions for Authors

The present review entitled “Role of Necroptosis in Intervertebral Disc Degeneration 2” by Md Abdul Khaleque and colleagues is well written and provide detailed description of the necroptosis machinery. The authors provide clear illustration of the complex signaling pathways. It is interesting that the author includes RIPK1 independent and dependent results for the same condition. I however have one major comment. While the manuscript is well written and informative many other reviews have focused on necroptosis from a molecular biology aspect. The premise of the review is about necroptosis in intervertebral disc degeneration, and this seems to be only in the background of the text while it should be the focus.

Bellow are suggestions to improve the present manuscript.

The original work by Degterev et al with necrostatin was more about RIPK1 than RIPK3

There is no alpha to TNF

Please check the spacing in the text

Please be consistent with the naming RIP or RIPK and with the drugs nomenclature GSK or GSK’ or post transcriptional modification p-RIP or P-RIP

Reference(s) should be provided for the HSP90 inhibitors, 4-PBA

The relevance of H2O2 as a model should be discussed.

A description of the different compression model, in vitro and in vivo, and how those may lead to necroptosis would strengthen the review

Schematic of necroptosis in the context of IVDD in vivo and in vitro would improve the present manuscript

Author Response

I am sharing an attachment in response to the reviewer's comments here.

Reviewer 2 Report

Comments and Suggestions for Authors

The submitted manuscript/review is focused on roles and regulation of necroptosis. The review is interesting, although it requires a lot of work. The paper should be edited by professional English editors. The review can be extended.

1.       There are many mistakes in English grammar and style. For instance in the Abstract: this mistake should be corrected ; “Necroptosis is triggers by…” – this should be “Necroptosis is triggered by…”. There are more mistakes in the text.

2.       Figures are not ready for publication. Very low quality. Authors should use specialised software to design and prepare figures. Drawing of mitochondrion is very poor.

3.       Abstract should reflect the content. The review is focused on the compression. However, it was not presented in the Abstract.

4.       The review does not provide comprehensive description of necroptosis-targeting drugs which can be used for IVDD treatment. The section should also present a table with the most promising or potentially promising drugs (specific inhibitors like TUDCA) which can be -repurposed or developed for IVDD.

5.       The organization of sub-section should be improved. The potential treatment can be presented in a separate section before the Conclusions.

6.       The reference section can be extended. Authors should add more recent papers form 2022-2023.

Comments on the Quality of English Language

English style/grammar should be improved/corrected.

Author Response

I have attached a file as a response to the reviewer's comments. 

Reviewer 3 Report

Comments and Suggestions for Authors

The practical implications of this paper are that necroptosis plays a significant role in intervertebral disc degeneration, and targeting necroptosis may be a potential therapeutic strategy for treating this condition. The review highlights potential therapeutic targets for regulating necroptosis, including RIPK1, RIPK3, and MLKL. By understanding the molecular mechanisms of necroptosis and its role in intervertebral disc degeneration, researchers and clinicians may be able to develop new treatments that can slow or prevent the progression of this condition. However, further research is needed to fully understand the potential of necroptosis as a therapeutic target and to develop effective treatments based on this knowledge.

Regarding future works, the paper suggests several areas for further research, including:

- Investigating the role of necroptosis in other pathological conditions beyond intervertebral disc degeneration.

- Developing more specific and effective inhibitors of necroptosis-associated proteins, such as RIPK1, RIPK3, and MLKL.

- Studying the potential of combination therapies that target multiple pathways involved in intervertebral disc degeneration, including necroptosis, apoptosis, and inflammation.

- Conducting more studies in human subjects to validate the findings from animal models and in vitro studies. - Exploring the potential of biomarkers associated with necroptosis as diagnostic or prognostic tools for intervertebral disc degeneration.

Author Response

Thank you for your valuable comments

Round 2

Reviewer 1 Report

Comments and Suggestions for Authors

The authors have answered some of my questions and concerns, i have no additional suggestions. While it is a detail, it is unfortunate that the tumor necrosis factor remains mislabeled in the manuscript (JAMA Dermatol. 2016;152(5):557. doi:10.1001/jamadermatol.2015.4322).